# Factors that influenced utilization of antenatal and immunization services in two local government areas in The Gambia during COVID-19: An interview-based qualitative study

Abdourahman Bah [ORCID]*, Giuliano Russo [ORCID]

The Wolfson Institute of Population Health, Queen Mary University of London, London, United Kingdom

* abdourahmanbah93@gmail.com

**Data Availability Statement:** The dataset has been submitted with the manuscript

## Abstract

### Introduction

Evidence is being consolidated that shows that the utilization of antenatal and immunization services has declined in low-income countries (LICs) during the COVID-19 pandemic. Very little is known about the effects of the pandemic on antenatal and immunization service utilization in The Gambia. We set out to explore the COVID-19-related factors affecting the utilization of antenatal and immunization services in two Local Government Areas (LGAs) in The Gambia.

### Methods

A qualitative methodology was used to explore patients' and providers' experiences of antenatal and immunization services during the pandemic in two LGAs in The Gambia. Thirty-one study participants were recruited from four health facilities, applying a theory-driven sampling framework, including health workers as well as female patients. Qualitative evidence was collected through theory-driven semi-structured interviews, and was recorded, translated into English, transcribed, and analysed thematically, applying a social-ecological framework.

### Results

In our interviews, we identified themes at five different levels: individual, interpersonal, community, institutional and policy factors. Individual factors revolved around patients' fear of being infected in the facilities, and of being quarantined, and their anxiety about passing on infections to family members. Interpersonal factors involved the reluctance of partners and family members, as well as perceived negligence and disrespect by health workers. Community factors included misinformation within the community and mistrust of vaccines. Institutional factors included the shortage of health workers, closures of health facilities, and the lack of personal protective equipment (PPEs) and essential medicines. Finally, policy

**Funding:** The author(s) received no specific funding for this work.

**Competing interests:** The authors have declared that no competing interests exist.

factors revolved around the consequences of COVID-19 prevention measures, particularly the shortage of transport options and mandatory wearing of face masks.

## Conclusions

Our findings suggest that patients' fears of contagion, perceptions of poor treatment in the health system, and a general anxiety around the imposing of prevention measures, undermined the uptake of services. In future emergencies, the government in The Gambia, and governments in other LICs, will need to consider the unintended consequences of epidemic control measures on the uptake of antenatal and immunization services.

## 1. Introduction and background

With the ongoing COVID-19 pandemic, the achievements made globally in improving mother, newborn and child health (MNCH) have been threatened. During the early phase of the pandemic, there was a concern about the potential direct impact of COVID-19 on vulnerable populations, particularly women and children [1]. The direct impacts that were of particular concern included potential increased maternal mortality, prematurity, stillbirths and congenital birth defects due to transmission from mothers to newborns [2–4]. Evidence gathered thus far on the direct impacts of COVID-19 on MNCH indicates a substantial increase in maternal, newborn and child mortality, particularly in LICs [5].

Given the introduction of strict COVID-19 prevention measures, such as lockdowns and other movement restrictions, it became clear that there would be some indirect effects resulting from these measures, particularly in LICs [6, 7]. In fact, studies conducted thus far have reported disruptions in the usage and provision of antenatal and immunization services during the pandemic resulting from COVID-19 prevention measures [8–12]. For example, a study conducted across 18 LICs estimated an average 2.6% to 4.6% decline in antenatal and immunization service utilization during the pandemic [13].

Evidence on the COVID-19 factors associated with the disruptions suggests that a combination of demand-side factors (i.e., factors related to antenatal and immunization service users) and supply-side factors (i.e., those related to the health system) are responsible for the disruptions. The demand-side factors include movement restrictions, economic hardship and fear of getting infected [14–17]. The supply-side factors, on the other hand, include health workforce difficulties, unavailability of services, shortage of supplies and suspension of services [18–20]. These findings are in line with the findings of other studies conducted in Liberia, Sierre Leone and Guinea during the 2013–2016 Ebola outbreak that showed that fear of infection, transport difficulties, closure of essential health services and the attitudes of care providers were the main deterrents to antenatal and immunization service utilization during the Ebola outbreak [21–25].

In The Gambia, the uptake of MNCH services has improved significantly over the last decade. For example, the proportion of women that delivered in health facilities increased from 63% in 2013 to 84% in 2019; the percentage of women that received a postnatal check-up within two days of delivery increased from 76% in 2013 to 88% in 2019; and about 79% of women made at least four antenatal care visits and about 98% of women received antenatal care from a skilled health professional (doctor, nurse or midwife) in 2019 (Table 1) [26, 27]. In addition, the country has recorded improvements in vaccination coverage for children between 12 and 23 months of age (Table 1). For example, the basic vaccination coverage increased from 76% in 2013 to 85% in 2019 [26, 27].

**Table 1. Key maternal and child health indicators in The Gambia in 2019.**

| Maternal health | Percentage |
|---|---|
| Births delivered in a health facility (%) | 84 |
| Birth assisted by a skilled provider (%) | 84 |
| Postnatal care for mothers within two days of delivery (%) | 88 |
| Antenatal care–four or more visits | 79 |
| Maternal mortality (deaths per 100,000 live births) | 289 |
| **Child health** | |
| Children who have received all basic vaccinations (%) | 85 |
| Children who have received all age-appropriate vaccinations (%) | 77 |
| Infant mortality rate (deaths per 1,000 live births) | 42 |
| Under-five mortality rate (deaths per 1,000 live births) | 56 |
| Neonatal mortality rate (deaths per 1,000 live births) | 26 |

Source: Gambia Bureau of Statistic (GBoS) and ICF International [26, 27]

However, despite these achievements in increased MNCH uptake, the maternal, newborn and child mortality ratios in The Gambia remain very high (Table 1). The country has an under-five mortality ratio of 56 per 1,000 live births, an infant mortality ratio of 42 per 1,000 live births and a neonatal mortality ratio of 29 per 1,000 live births [26]. Concerning maternal deaths, the maternal mortality ratio is 289 per 100,000 live births [26]. These high mortality ratios are attributable to the following factors: inadequate financial and logistical support; a shortage of medical supplies and equipment; worsening physical infrastructure; inadequate numbers of health workers; high attrition rates; and an inadequate referral system (among other factors) [28].

The ongoing COVID-19 pandemic threatens to reverse the progress made so far in improving the uptake of antenatal and immunization services in The Gambia. The first confirmed case of COVID-19 was detected in The Gambia on 17 March 2020 and 3 more cases were detected by the end of March (Table 2) [29, 30]. However, by the end of June 2020, only 45 new confirmed cases had been detected [31]. Nevertheless, from July to September 2020, the country experienced a short and intense first wave of infections, during which it recorded 3,530 new confirmed cases and 110 deaths [32]. This peak period was followed by a period of low infections till December 2020, after which a second wave commenced in January 2021 and ended in March 2021, with a total of 5,459 confirmed cases and 165 deaths [33, 34]. This period was followed by a slow period until June 2021 [35] and a third wave then began in July 2021 and continued until the end of September 2021. The country recorded a total of 3,856 new confirmed cases during this period (Table 2) [36].

Shortly after the first COVID-19 case was detected in The Gambia on 17 March 2020, the government imposed a national lockdown and closed its international land, sea and air borders [37]. On 27 March, the country declared a state of emergency, which mandated the closing of schools, non-essential shops, and places of worship, the prohibition of social gatherings of more than 10 people, and the limiting of the number of passengers on public transport [38]. Between April and July 2020, the government introduced contact tracing and quarantine measures which obliged suspected and confirmed cases to remain in hotels for 14 days. As the pandemic progressed, hotel quarantine was replaced with self-isolation at home for both suspected and confirmed cases for a period of 10 days [37]. From July 2020, the government made the wearing of face masks compulsory in public, and introduced curfews and social distancing measures in public, including in health facilities and on public transport [39].

**Table 2. COVID-19 cases in The Gambia over time.**

| Months | New confirmed cases | Total numbers of confirmed cases | New confirmed deaths | Total numbers of confirmed deaths |
|---|---|---|---|---|
| March 2020 | 4 | 4 | 1 | 1 |
| April to June 2020 | 45 | 49 | 1 | 2 |
| July to September 2020 | 3530 | 3579 | 110 | 112 |
| October to December 2020 | 218 | 3797 | 12 | 124 |
| January to March 2021 | 1662 | 5459 | 41 | 165 |
| April to June 2021 | 620 | 6079 | 16 | 181 |
| July to September 2021 | 3856 | 9935 | 157 | 338 |

Source: Ministry of Health [31–36]

The COVID-19 pandemic and its prevention measures may have affected the utilization of antenatal and immunization services in The Gambia. In fact, evidence collected thus far suggests that the uptake of antenatal and immunization services declined considerably during the first wave of the pandemic, which ended in September 2020 [6, 12]. However, the factors responsible for this decline have so far been unexplored. To fill this gap, this study therefore investigated the COVID-19-related factors associated with a decline in the uptake of antenatal and immunization services in two LGAs in The Gambia.

## 2. Materials and methods

### 2.1 The methodological approach

The study utilised an exploratory qualitative approach to explore the COVID-19-related that influenced the uptake of antenatal and immunization services. This methodology was chosen as it allows the exploration of participants' perceptions and experiences in-depth, leading to a more comprehensive understanding of the COVID-19-related factors influencing the uptake of antenatal and immunization services, which would not be possible with a quantitative approach [40]. The social-ecological framework was used to explore and understand the COVID-19-related factors influencing the uptake of antenatal and immunization services. This framework was chosen because it provides a theory-based framework for understanding the dynamic interrelations between various factors that shape individual behaviour [41]. The framework illustrates five levels of influence on individual behaviour: individual, interpersonal, community, institutional and policy. The framework guided the preparation of the interview guides and also provided an organisational framework for data analysis. It was also useful in producing themes and sub-themes that take into account the wide range of factors, including demand-side and supply-side factors, that influenced the use of antenatal and immunization services in the period being studied.

The 21-item checklist from the Standards for Reporting Qualitative Research (SRQR) was used to inform the qualitative approach of the research and to report on the interview findings [42]. Scientific and ethical approvals were obtained from the Medical Research Council Unit of The Gambia (MRCG) Scientific Coordinating Committee and The Gambia Government/ MRCG Joint Ethics Committee, respectively. Official permission letters were also obtained from the Director of Health Services at the Ministry of Health and from the administrative offices of all four health facilities. After explaining the purpose of the study, written consent was obtained from all study participants. To ensure confidentiality and anonymity, the interview data were de-identified before analysis. This was done by assigning each study participant a unique identification code.

## 2.2 Data collection

The study was conducted from June 2021 to August 2021 in Brikama and Kanifing LGAs, the two most densely populated LGAs in The Gambia, home to about 37% and 20% of The Gambia's population, respectively [27]. Both areas are situated close to the Atlantic coast, and mostly comprise urban areas. These two areas were purposively selected based on Ministry of Health reports on the incidence of COVID-19 cases. They were reported to have the highest incidence of confirmed COVID-19 cases in the country [43]. Also, the decision to restrict the study to these two areas was determined by feasibility issues relating to the limited timeframe for the study and logistical challenges. Consequently, four health facilities were purposively selected from these two areas, to represent both public and private health facilities (see Table 3 below).

Seventeen mothers were purposively selected from the above-mentioned health facilities. A convenience sampling strategy was used to select mothers that regularly accessed antenatal and immunization services during the pandemic and mothers that did not attend all the recommended immunization and antenatal visits [44]. In addition to these inclusion criteria, only mothers who were able to provide informed consent, were at least 18 years of age, and who had a child not older than one-year-old were considered for inclusion in the study. The selected participants also had varying socioeconomic statuses, ages and education levels. The mothers that met the inclusion criteria were identified with the help of the health workers using maternity registers and child vaccination cards. In addition to the mothers selected, 14 healthcare providers were purposively selected from the above-mentioned health facilities. They included nurses, midwives and allied health professionals who were directly involved in providing antenatal and immunization services during the pandemic. The health providers were selected to explore their perspectives on the COVID-19 factors shaping access to antenatal and immunization services.

Data were collected through semi-structured interviews. A total of 31 interviews were conducted among the two groups, comprising 14 health workers and 17 mothers. All interviews were conducted in the health facilities. The interviews were conducted using topic guides created based on the findings of the literature review (see S1 and S2 Files). The topic guides were designed specifically for each group and each interview guide was divided into the following five areas of inquiry, which are based on the five levels of the social-ecological framework: individual, interpersonal, community, institutional and policy factors. The interview guides were piloted with two participants. All interviews were conducted in English, Mandinka and Fula.

**Table 3. List of people interviewed by participant category, LGA and health facility.**

| Type of interviewee | LGAs | | | | Total |
|---|---|---|---|---|---|
| | Kanifing LGA | | Brikama LGA | | |
| | Public health facility | Public health facility | Public health facility | Private health facility | |
| | Kanifing General Hospital (KGH) | Bundung Maternity and Child Health Hospital (BMCHH) | Brikama District Hospital (BDH) | Africmed International Hospital | |
| Midwife | 3 | 3 | - | - | 6 |
| Nurse | - | 2 | - | - | 2 |
| Allied health professional | - | 3 | 2 | 1 | 6 |
| **Total of health workers interviewed** | **3** | **8** | **2** | **1** | **14** |
| Mothers that regularly accessed antenatal and immunization services | - | 5 | 2 | 1 | 8 |
| Mothers that did not regularly access antenatal and immunization services | - | 6 | 2 | 1 | 9 |
| **Total of mothers interviewed** | **-** | **11** | **4** | **2** | **17** |

All the interviews were conducted by the first author, who is fluent in all the three languages. Each interview lasted between 15 to 30 minutes. The interviews were audio-recorded and, where necessary, translated into English, and were transcribed verbatim and anonymised.

## 2.3 Data analysis

The data collected from the interviews were analysed using a thematic approach. The analysis was conducted in several stages. To obtain a first impression of the data, the researchers read through the transcripts to familiarise themselves with the content. Following this, the transcribed data were entered into NVivo 12, and a deductive approach was used to identify predetermined codes based on the topic guides and literature review. An inductive approach was then used to identify new codes that were different from the predetermined codes. Once all the data had been coded, similar codes from mothers and health workers were triangulated and combined into themes and sub-themes based on the social-ecological framework and the research objectives. To ensure the objectivity of the findings, both researchers coded everything and compared the codes that emerged and worked closely together in reviewing and defining themes. This procedure was repeated until no new codes emerged from the data. Finally, all of the coded data were reviewed to ensure that all the codes fit their respective themes and the findings were then reported using individual quotes.

# 3. Results

In this section, we lay out the main findings from our qualitative interviews, according to the social-ecological model. First, we present the socio-demographic characteristics of the study participants, then we present the themes that emerged from the interviews.

## 3.1 Socio-demographic characteristics

A total of 17 mothers participated in this study. The ages of the mothers ranged from 18 to 35. Sixteen of them were married, while only one was single. Six of them did not have any formal education while 11 reported to have attended at least primary education. The majority of them belonged to the Mandinka (eight) and Fula (five) ethnic groups, while the rest belonged to the Jola (two), Wollof (one) and Manjago (one) ethnic groups. Regarding their religious beliefs, 14 of the mothers were Muslims, while three were Christians. In addition to the mothers interviewed, 14 healthcare workers were interviewed. Out of these 14, six were midwives, two were nurses and six were allied health professionals.

## 3.2 Individual factors

This category of factors refers to the demand-side factors at the individual level that limit women's uptake of antenatal and immunization services. They include anxiety arising from fear of getting infected with COVID-19, fear of being quarantined and fear of infecting family members.

**3.2.1 Fear of getting infected.**   Both health workers and mothers mentioned that fear of getting infected was a contributing factor to the reduction in the uptake of antenatal and immunization services. Mothers perceived health facilities as a source of infection, due to overcrowding in health facilities and the lack of proper ventilation, which increases the risk of infection. They also mentioned a lack of social distancing in health facilities due to the lack of sufficient space in health facilities, the lack of enough seats, and some facilities failing to observe COVID-19 precautionary measures, such as wearing face masks and ensuring hand

hygiene. This led some mothers to feel it would be unsafe to come to health facilities for antenatal and immunization services.

*"During the peak of the pandemic, I was pregnant, but I was not coming for antenatal service until after six months of my pregnancy . . . During the pandemic, I was not coming to the health facility that often. This was because of fear of getting infected." (Respondent 1)*

*"Looking at the little space available in many facilities, it was nearly impossible to practice social distancing. This contributed to people not going to health facilities since it is a gathering." (Health worker 14)*

*"It was also not safe to come to the health facility because you can easily get infected at the health facility. I wasn't coming regularly because the health facility used to be overcrowded, which made the environment unsafe." (Respondent 13)*

**3.2.2 Fear of being quarantined.** Other mothers mentioned that testing positive for COVID-19 and being quarantined was another factor that influenced the utilization of antenatal and immunization services. They were worried that if they went to the health facilities and were screened for COVID-19 through temperature checks, they would be told that they had COVID-19 even if they were not actually infected, and this would mean being quarantined in a government facility for two weeks. This was a huge worry for them as it would mean being away from their loved ones and potentially being stigmatised within their communities.

*"I remember there was a time when you tell someone you should go to a health facility. . . . they would tell you I will not because if I go to the health facility, they will tell me I have COVID-19 and I will be quarantined." (Health worker 12)*

*"I have seen many of them not going to the health facility during the pandemic. The reason they would give is that if they go the health facility, they would be told that they have COVID-19. So, for that reason, they are afraid of going to the health facility." (Respondent 2)*

*"In my community, people were not going to health facilities that much. They used to say that if they go to the health facility, they will be tested for COVID-19 and they would be told that they have COVID-19, especially if you are having a cold. They would say that if you go there coughing and sneezing, they will say that you have COVID-19." (Health worker 11)*

**3.2.3 Fear of infecting family members.** Some mothers also reported being afraid of infecting family members. They were worried that if they went to the health facilities, they would get infected and would eventually infect their family members. This was particularly worrying for those living with vulnerable people, such as the elderly and those with compromised health status.

*"There were several people in my community who stopped taking their children for immunization during the pandemic. . . . They would say that they would not take their children for immunization to the health facility because the health facility is not safe, and they could easily get infected and bring it home to infect their families." (Respondent 4)*

*"We were always worried that we will get infected and take the disease to our families because some patients were not willing to follow the precautionary measures." (Health worker 8)*

### 3.3 Interpersonal factors

This category of factors refers to a combination of demand- and supply-side factors related to the interrelationships between the participants, their partners, family members and health workers.

**3.3.1 Husband's and family's attitude.** Some mothers reported that their partners refused to give them permission to access immunization and antenatal services during the pandemic. Therefore, to avoid potential conflict with their partners, some mothers decided against attending antenatal and immunization services. Others also mentioned that their family members, including their parents and their partner's family, dissuaded them from coming for immunization and antenatal services due to fear of infection and high transportation costs.

*"There was a reduction in the inflow and then the majority of MCH users when we called, they were telling us that because of the pandemic our family members are not allowing us to move out or go to health facilities, where it was always crowded." (Health worker 14)*

*". . .there were some members of my family who were telling me not to go to the health facility because of the COVID-19 pandemic." (Respondent 3)*

*"Some of my family members were saying I should not come because of the high transport fares that I have to pay to get here. Some were also saying I shouldn't come because the health facility was not safe during the pandemic." (Respondent 13)*

**3.3.2 Relationship with health workers.** Some mothers cited mistreatment and disrespect from health workers as being a deterrent to coming for immunization and antenatal services during the pandemic. They mentioned that they stopped coming to health facilities because of the bad behaviour exhibited by some health workers, including being disrespectful and even insulting them. The reason that the mothers gave for this behaviour was the fact that the health workers were short-staffed during the pandemic and were always exhausted. It was reported that this resulted in frustration, and that this frustration was vented on patients. Some mothers also reported they were neglected by the health workers. They mentioned that they did not receive services in a timely manner at health facilities during the pandemic as the health workers were demotivated and, as such, neglected their duties, including sitting around and chatting with their colleagues instead of providing services. As such, some mothers mentioned that they had frequent clashes with the health workers. This was corroborated by several health workers. They confirmed that some health workers refused to provide antenatal and immunization services because, even though they were sacrificing their lives, they felt they were not adequately supported by the government. For instance, they mentioned that they were not provided with incentives to motivate them. As a result, some of the health workers were demotivated and refused to provide antenatal and immunization services.

*"It might be one of the reasons why some people were not bringing their children for immunization because for me, personally, if you embarrass me in front of people, I will stop coming to the hospital. . . . Sometimes, people even choose to go to the big pharmacies or private hospitals, where they will not be treated badly by the health workers." (Respondent 6)*

*"I always quarrel with them because you come from a far place. Instead of them attending to you, they will be sitting and chatting with their colleagues and neglecting you." (Respondent 2)*

*"The quality of service was very much affected, because, during those days, some of our staff were refusing to provide the service." (Health worker 1)*

### 3.4 Community factors

This category of factors refers to those demand-side factors at the community level that reduced mothers' willingness to attend both antenatal and immunization services. These factors include vaccine scepticism and narratives about COVID-19 within the community.

**3.4.1 Mistrust in vaccines.**   The research participants described having a mistrust of vaccines since the COVID-19 outbreak, often citing pervasive rumours that health workers were injecting patients with the COVID-19 virus. Participants also mentioned that within the community childhood vaccines and the COVID-19 vaccine were sometimes confounded. As a result, some mothers refused to come for antenatal services and others refused to bring their children for immunization because they believed that if they came to health facilities, they would be given the COVID-19 vaccine instead of maternal and child vaccines.

*"They would not even trust the childhood vaccines we were giving them here. The perception they had was that we were giving them COVID-19. They believed that the COVID-19 virus was included in the childhood vaccines." (Health worker 2)*

*"Some mothers would not take their children for immunization because they thought that if they bring their children for immunization, they will be given the Covid-19 vaccine which is not safe." (Respondent 3)*

*"Some [mothers] also started to confuse the vaccines we give them with COVID-19 vaccines. Some husbands were telling their wives not to take our childhood vaccines. As a result, the number of patients coming here was very reduced." (Health worker 4)*

**3.4.2 Misleading information.**   Both health workers and mothers cited misleading information and rumours about vaccines arising from the media as another factor limiting mothers' willingness to access antenatal and immunization services. They mentioned that people used social media and other forms of communication to advise people not to visit health facilities and not to get their children immunised. Others mentioned that they did not come for antenatal and immunization services because there were rumours within their community that it was not safe to visit health facilities. Some health workers described their encounters with women who did not come for antenatal and immunization services: they reported that the mothers were erroneously told in their communities that health facilities had been closed and the provision of antenatal and immunization services had been suspended.

*"The rumours from the community and from the media also contributed to my unwillingness to go to the health facility because they were advising people in the media not to go to the health facility as it is not safe." (Respondent 1)*

*"Media issues are disturbing us and vaccine hesitancy. People tend to write information anyhow they feel. Some people believe the media more than the health workers. That was affecting us, . . .they keep telling people not to send their children to health facilities and not to get them immunised. Even here, some would bring their children, but the moment they know we are going to immunise them, they refuse." (Health worker 4)*

*"Others would even tell you that health clinics [antenatal and immunization services] have stopped. We don't even know where they got that information. They would tell you that they have heard that the hospital has stopped offering immunization services. So, for that reason, they were no longer taking their children for immunization." (Health worker 2)*

### 3.5 Institutional factors

This category of factors refers to the main supply-side (i.e., health system) factors that contributed to the decline in the uptake of antenatal and immunization services during the COVID-19 pandemic. These factors include shortages of health workers, a lack of essential materials, including personal protective equipment (PPE) and medicines, and the closure of some health facilities.

**3.5.1 Inadequate personal protective equipment.** Several mothers reported that some health workers did not use PPE. For instance, they mentioned that some health workers did not wear face masks. As a result, some mothers were unwilling to come for antenatal and immunization services as they felt it would be unsafe. For their part, the health workers blamed this on the lack of PPE. For example, they reported acute shortages of face masks and gloves.

*"As some health workers do not follow the precautionary measures . . . the government should do something about that. Because that was the main reason, I was not coming to the health facilities." (Respondent 1)*

*"The main barrier we had here is the lack of PPE, such as face masks and gloves." (Health worker 13)*

**3.5.2 Shortage of essential medicines.** Some mothers reported a lack of essential medicines in the health facilities during the pandemic. They reported that whenever they went to the health facility, they would only receive a few of the medicines they needed; the rest they had to buy them in the pharmacies, which was reported to be quite expensive. As a result, some of them stopped going to health facilities altogether.

*"The pandemic made things very difficult because there were not even enough medicines at the health facilities. You would go to the health facility, and they would ask you to go and buy the medicines from the private pharmacies. . . . Some mothers are not willing to take their children to the health facility because they know that they will not have the medicines they need." (Respondent 2)*

*"Sometimes even I.V fluid, Zinc syrup and vitamin C were scarce. . . .The shortage of medical supplies could have a been reason why some people were not coming to health facilities because for example, if I leave my home and come here and don't have the medicines that I need, I will go home and not come back here in the short distant future." (Health worker 1)*

**3.5.3 Staff shortages.** Some mothers complained about a shortage of health workers during the pandemic. They reported that this shortage resulted in an increase in waiting times for both antenatal and immunization services as the provision of these services was slower than usual. This deterred some mothers from going to health facilities, as they were worried that the longer they remained in the health facility the greater their chance of getting infected. This claim was corroborated by most health workers as they described an acute shortage of health

workers during the pandemic. According to them, this shortage was a result of staff getting infected and the deployment of staff towards the prevention and control of COVID-19. Some reported that this shortage led to an increased workload and to staff burnout.

*"During the pandemic, there were not enough health workers at this health facility because many of them were not working at that time. If there used to be three health workers on duty before the pandemic, this was reduced to two health workers during the pandemic. So, this resulted in increased waiting time for us as the service was a bit slow because there were not enough health workers." (Respondent 4)*

*"Yes, during the pandemic, we had a shortage of manpower because some of our colleagues were sick, and others were diagnosed with COVID-19 and had to quarantine for two weeks, along with those who came in contact with them. This affected our service delivery. The work-load also increased for those of us who remained because we had to cover for those who were absent. This made our work very difficult at that time." (Health worker 7)*

**3.5.4 Scaling down of antenatal and immunization service provision.** Some mothers reported that they stopped attending antenatal and immunization services because of the clo-sure of some health facilities. They reported that they were sent home several times from health facilities. Others reported that some health facilities temporarily suspended the provision of immunization and antenatal services during the pandemic. Some mothers also reported that in order to reduce overcrowding, some health facilities introduced a limit on the number of people allowed to receive immunization and antenatal services each day. Thus, to receive these services, mothers had to come to the health facility very early, which was not possible for most of them. As a result, some stopped attending antenatal and immunization services altogether.

*"I used to go to Hagan Hospital in Banjul, but when the COVID-19 pandemic started, they closed the health facility. I brought my child there for immunization, but they said they have closed the hospital and they are not seeing patients. I went there several times, but they told me that it was closed. So, I went home and stopped taking my child for immunization." (Respondent 2)*

*"I was not taking him for immunization every month. . . .At that time, you only take your child for immunization when you have an appointment, which was about every two months. This was because they stopped weighing children during the pandemic. So, I used to bring him only when he was supposed to get vaccinated." (Respondent 11)*

*"They also introduced a limit on the number of people they would allow in the health facility a day, so if you don't come early, you will not receive the service. For this reason, many people stopped going to the health facility. Even myself, there was a time I stopped going to the health facility." (Respondent 14)*

## 3.6 Policy factors

This category of factors refers to infection prevention measures introduced during the COVID-19 pandemic that affected the uptake of antenatal and immunization services. These include transportation challenges arising from movement restrictions and face mask enforce-ment issues.

**3.6.1 Movement restrictions.**   Some mothers reported that the movement restrictions imposed by the government resulted in transportation challenges that affected their ability to access antenatal and immunization services during the pandemic. They reported an acute shortage of vehicles as some of the drivers were reluctant to work because of the social distancing measures and movement restrictions, which made it particularly difficult to travel to health facilities. Several mothers also reported high transportation costs being a factor that influenced the utilization of antenatal and immunization services during the pandemic. They reported there was a substantial increase in transport fares and reported that this posed a significant challenge for most of them as they could not afford to pay the high transport costs due to the economic constraints caused by the pandemic.

*"Transport was a big problem for me during the pandemic because I live all the way in Lamin, which is quite far from here. There was a shortage of vehicles as many drivers were not working at that time due to social distancing measures introduced by the government." (Respondent 4)*

*"Of course, I had transport difficulties at that time because, at that time, I was staying in Bundung. I used to pay about 50 dalasis to get to Banjul. When coming back, I used to face the same problem. Transport at that time was very costly and there were not many vehicles. Their decision to stop us from going to the health facility was a huge favour to me because I used to spend a lot of money on transport." (Respondent 7)*

*"In my village, there were some people who were not coming for antenatal and immunization services during the pandemic. They were not coming to the health facility because of high transport fares." (Respondent 13)*

**3.6.2 Face mask enforcement.**   Some mothers complained about the fact that the wearing of face masks in health facilities and on public transport was mandatory. They reported that people without face masks were not allowed entry into the health facilities and were denied access to antenatal and immunization services. Others complained about the discomfort associated with the mandatory wearing of face masks. This was particularly the case for those with breathing problems, such as asthma patients. Some also reported PPE costs being a factor that influenced the utilization of antenatal and immunization services during the pandemic. They reported that the cost of face masks was quite high, and most of them could not afford this.

*"Others would sometimes be sent home because they don't have face masks and their children would not be immunised." (Respondent 4)*

*". . .we did not allow people to access MCH services without being in a protective mask. you had to put on a face mask before you can access MCH services here. We went even to the extent that we sent people out if they don't put on a face mask" (Health worker 5)*

*"The mandatory wearing of a face mask was another reason why I was not willing to go to health facilities because I feel uncomfortable wearing it. . . . When I put it on, I feel like suffocating." (Respondent 1)*

*"Sometimes, I used to see people not coming to the health facility because they don't have the money to buy a face mask." (Respondent 1)*

## 4. Discussion

This study seeks to contribute to the qualitative literature exploring COVID-19-related factors that influenced the uptake of antenatal and immunization services in urban areas of The Gambia during the pandemic. Based on the social-ecological model, we identified several factors that influenced the use of antenatal and immunization services in two LGAs in The Gambia during the pandemic: anxiety and fear of getting infected in health facilities, negative attitudes of healthcare workers, a mistrust of the health system and of vaccines, shortages of health workers, perceptions that there was inadequate equipment and there were inadequate essential medicines, and barriers imposed by travel restrictions and the mandating of wearing face masks. The factors identified in this study are not unique to The Gambia.

This is one of the first studies to document the reality of the COVID-19 pandemic in the comparatively less researched country of The Gambia. Some of our findings are consistent with similar work conducted in other countries in sub-Saharan Africa. Studies from Ethiopia and Nigeria, for example, reported that anxiety resulting from fear of testing positive for COVID-19, uncertainty and stress about the pandemic prevented mothers from accessing immunization and antenatal services [15, 18]. Similarly, a study conducted in India found that mothers were unwilling to seek antenatal and immunization services at health facilities due to fear of contracting COVID-19 and transmitting it to their babies [9].

Our analysis of the importance of interpersonal factors for the uptake of health services is corroborated by the literature reporting negative attitudes of health workers and low motivation to provide antenatal and immunization services, possibly in connection with low pay and high workloads [45]. Hailemariam and colleagues [18] found that mothers were unwilling to seek antenatal services due to fear of mistreatment by health workers and poor quality of care arising from lack of motivation of health workers to provide antenatal services.

The theme of vaccine hesitancy and misinformation has been addressed before: vaccine mistrust and misinformation emanating from the media and communities likewise prevented mothers from seeking immunization and antenatal services during the Ebola outbreak in West Africa [21]. However, in comparison to the Ebola epidemic we found a larger role played by social media in sharing misinformation during the present pandemic, possibly because of the spread of mobile technology. COVID-19 also received comparatively greater media coverage in The Gambia, which may have given more visibility and spread to dubious sources and unofficial information in this West African country.

Many of our interviewees reported a lack of protective equipment in The Gambia's health facilities. According to the health workers we talked to, this fuelled an unwillingness among many to provide services, for fear of getting infected, the closure of some facilities, and the suspension of routine antenatal and immunization services. Such findings are consistent with previous studies that have showed how the pandemic disrupted the provision of immunization and antenatal services due to the deployment of health workers to COVID-19-related tasks and the disruption of global supply chains for medicines and PPEs [45, 46].

A few limitations need to be considered when interpreting our study findings. First, it was not possible to quantify the extent to which of the above factors influenced the utilization of antenatal and immunization services and was not possible to rule out the existence of some of these factors before the pandemic and their relevance during other outbreaks like Ebola. Second, the distinction between demand-side and supply-side factors was not always clear-cut in our interviews, as both sets of factors are known to interact in health markets [47]. Third, the study was conducted between June 2021 and August 2021 –a year after the first wave of COVID-19 in The Gambia and about six months after the second wave. Thus, given the fact that the study participants were required to recall events that occurred in the past six to 12

months, there may have been some recall bias. Fourth, there could also be the issue of selection bias as the pool of patients was drawn from health facilities and those who did not seek care were not included. The authors also acknowledge that there could be a possible social desirability bias given that the interviews occurred inside health facilities. Also, due to difficulties in getting enough participants, only 17 mothers were interviewed mostly from the 2 public hospitals and hardly any from the private practice and some the study findings were the interviewees perspective of what was happening in their community and not their personal experience. Finally, the study was conducted in only two LGAs and was urban centric as the sample was from urban areas; as such, the study findings are not generalizable to The Gambia but provides insights for the country.

In our view, these findings have policy implications for the government in The Gambia and for governments in other LICs, for the current pandemic as well as for future epidemics. The fear of contracting COVID-19 and the vaccine scepticism uncovered in our study need to be addressed. This could be achieved by providing health education to mothers to help counter unfounded narratives about unsafe childhood vaccines and health facilities [48]. The reported fears of mistreatment by health workers are also worrisome, as this erodes trust in health institutions and jeopardises the gains made in immunization rates over the last two decades. There is no simple solution to this complex problem, but governments will need to balance the stick and carrot interventions at their disposal, by improving the financial and professional conditions of the workforce but also linking them to performance and patient satisfaction [49]. Ultimately, the uptake of newer and constantly evolving antenatal and immunization services will always depend on mothers' willingness to seek such services from health professionals that treat them with dignity and respect. Furthermore, our study used a socio-ecological framework to explore the demand-side and supply-side factors influencing the use of antenatal and immunization services during the pandemic in The Gambia: a similar approach might be employed to disentangle the multi-layered effects of the pandemic on health systems in other LICs. Because of the cultural, political and health systems similarities, we particularly hope that such an approach may be used to explore COVID-19 impacts in the African continent.

## 5. Conclusion

There are still gaps in our knowledge of the ways the COVID-19 pandemic is impacting the utilization of lifesaving antenatal and immunization services in LICs, particularly for small countries like The Gambia. We aimed to fill these gaps by conducting theory-driven semi-structured interviews exploring the factors that influenced the utilization of antenatal and immunization services in two LGAs in The Gambia with high COVID-19 prevalence.

We found that the prevention measures introduced during the pandemic negatively affected the utilization of antenatal and immunization services in The Gambia, particularly due to challenges relating to the mandatory wearing of face masks and movement restrictions. Mothers' anxiety and fear about contracting COVID-19 in health facilities, and the health system's inability to sustain the provision of antenatal and immunization services during the pandemic, also undermined the uptake of antenatal and immunization services.

Despite its limitations, this study has the merit of bringing a focus to the shortcomings of health services in The Gambia, and to the need for governments to pay attention to the demand-side factors affecting health interventions in LICs, as well as to the need to counter misinformation and mitigate fears of the public during health emergencies. Finally, to prepare for future health emergencies, the health facilities and the government should consider stockpiling extra PPEs and planning the supplies of essential medicines.

## Supporting information

**S1 File.**
(ZIP)

**S2 File. Inclusivity in global research.**
(DOCX)

## Author Contributions

**Conceptualization:** Abdourahman Bah, Giuliano Russo.

**Data curation:** Abdourahman Bah.

**Formal analysis:** Abdourahman Bah, Giuliano Russo.

**Investigation:** Abdourahman Bah.

**Methodology:** Abdourahman Bah, Giuliano Russo.

**Project administration:** Abdourahman Bah.

**Resources:** Abdourahman Bah.

**Software:** Abdourahman Bah.

**Supervision:** Abdourahman Bah, Giuliano Russo.

**Validation:** Abdourahman Bah, Giuliano Russo.

**Visualization:** Abdourahman Bah.

**Writing – original draft:** Abdourahman Bah, Giuliano Russo.

**Writing – review & editing:** Abdourahman Bah, Giuliano Russo.

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
