## [Decision Letter · Decision Letter 0]

31 Jan 2023

PONE-D-22-27435Barriers to accessing mother, new-born and child health services in urban Gambia during COVID-19: An interview-based qualitative study.PLOS ONE

Dear Dr. Bah,

Thank you for submitting your manuscript to PLOS ONE. After careful consideration, we feel that it has merit but does not fully meet PLOS ONE’s publication criteria as it currently stands. Therefore, we invite you to submit a revised version of the manuscript that addresses the points raised during the review process.

We look forward to receiving your revised manuscript.

Kind regards,

Ribka Amsalu

Academic Editor

PLOS ONE

and https://journals.plos.org/plosone/s/file?id=ba62/PLOSOne_formatting_sample_title_authors_affiliations.pdf.

2. You indicated that you had ethical approval for your study. Please clarify whether minors (participants under the age of 18 years) were included in this study. If yes, in your Methods section, please ensure you have also stated whether you obtained consent from parents or guardians of the minors included in the study or whether the research ethics committee or IRB specifically waived the need for their consent.

3. Please include a complete copy of PLOS’ questionnaire on inclusivity in global research in your revised manuscript. Our policy for research in this area aims to improve transparency in the reporting of research performed outside of researchers’ own country or community. The policy applies to researchers who have travelled to a different country to conduct research, research with Indigenous populations or their lands, and research on cultural artefacts. The questionnaire can also be requested at the journal’s discretion for any other submissions, even if these conditions are not met.  Please find more information on the policy and a link to download a blank copy of the questionnaire here: https://journals.plos.org/plosone/s/best-practices-in-research-reporting. Please upload a completed version of your questionnaire as Supporting Information when you resubmit your manuscript.

Additional Editor Comments:

Relevant study that provides an insight on the perspective of women and healthcare workers on the indirect impact of COVID-19 and the mitigation measures on availability and use of antenatal and immunization services in the Gambia. Major edit

Comments

1. Title: Barriers to accessing mother, new-born and child health services in urban Gambia during COVID-19: An interview-based qualitative study

- Be clear on the objective of the study and be consistent throughout from title till conclusion

MNCH vs antenatal and immunization, these terms are not interchangeable. For one, MNCH includes curative services.

- ‘New-born’ replace with ‘newborn’ throughout

- ‘barriers to accessing...’ or ‘factors that influenced utilization of…’

- “The Gambia” versus “the Gambia” versus “Gambia” – choose one and be consistent

2. Abstract

- Introduction: Good to avoid terms that are not supported by data “least developed…, least researched…” , low-income country with limited research on ..

- Results: ‘.. relating to factors that were responsible for the reduction in the uptake of antenatal and immunisation services’ since the study did not measure if there was reduction or not of antenatal and immunization service uptake, a better phrase could be ‘… factors that influenced uptake (use) of antenatal and immunization services...’

3. Introduction

4. Methods

Data collection:

- Sample size, why 17 mothers? Did you expect saturation by then? Most were from the 2 hospitals and not so much from the 3rd public hospital and hardly any from the private practice. Was that intended or were you unable to get participants? In either case – this should be reflected in the discussion as limitation, and reasons why .

- How did you classify mothers who regularly versus not regularly used services? Were they asked different questions? Were there unique perspective ..? were you trying to see change from their norm in terms of use of service?

- Who conducted the interviews (were the interviewers trained, male/female, health worker/not hw)? Possible social desirability bias given that the interview occurred inside a health facility

- How long did the interview take per person?

- Were the tool used for interview translated/back translated or ? dedicated translator or were the authors/interviewers fluent in all three languages ?

5. Results

- ‘claimed’ has negative undertone suggest using words like ‘reported’ throughout

- ‘Fear of being quarantined’ is this similar to being asked to stay at home and to social distance. Good to differentiate and clearly state.

- Given your small sample size per health facility, it is possible that the respondents might be identified by health facility name. We often advise to not include name of hospital or say hospital 1 or 2 to avoid possible identification of interviewee and breach of confidentiality.

- Some of the interviewees were describing what they perceive was going on in their community – good to say that in your discussion section some of your findings were also the interviewees perspective of what was going on in their community and not a personal experience.

o “There were several people in my community who stopped taking their children for immunisation during the pandemic..”.

- Reference “WhatsApp”

- “misleading information” or “rumors” – trying to understanding if the respondents defined the information as “misleading’ or “rumor” ? as some of the health facilities were closed as described under paragraph Scaling down of MNCH services.

6. Discussion

- Hard to follow flow of this section. Suggest restructuring: main/key finding, interpretation of your findings/reflection of findings as it relates to other African and non-African studies, implications of the findings, strength/limitations of study, recommendation (what could be done differently in the future based on your findings).

- First sentence

o Given the small sample and lack of generalizability in Gambia suggest editing to say gives insight in Urban areas of Gambia.

- Second sentence

o ‘Factors responsible for reduction’… suggest rewording… the study does not measure reduction but rather factors that influenced decision to use or not use services.

- Line 8

o “physical barriers” .. suggest to say barriers

- ‘small” country as compared to ?

- Limitations: I would also think these were limitations

o The small sample size , unclear if saturation was researched

o The sampling method, selection bias as the pool of patients was drawn from hospital and does not include those who did not seek care at all.

o Urban centric as sample was from urban areas.

o Hence, findings not generalizable to Gambia but provides insight for the country.

7. Figure 1. Missing title of Figure. Not sure it adds value. Is it trying to capture framework or key findings and needs language edits “mistreatment by ..”

Reviewers' comments:

Reviewer's Responses to Questions

**Comments to the Author**

1. Is the manuscript technically sound, and do the data support the conclusions?

Reviewer #1: Yes

2. Has the statistical analysis been performed appropriately and rigorously? 

Reviewer #1: Yes

3. Have the authors made all data underlying the findings in their manuscript fully available?

Reviewer #1: Yes

4. Is the manuscript presented in an intelligible fashion and written in standard English?

Reviewer #1: Yes

5. Review Comments to the Author

Reviewer #1: This study qualitatively assesses barriers to MNCH in two regions of the Gambia during the COVID-19 pandemic through summer 2021. Participants provide both current and former barriers. This study may be of great interest to facilities and governments preparing for future surges of COVID-19 and new pandemics as well as facilities trying to establish pre-COVID-19 levels of care. In addition to the Gambia, findings of this study may be relevant to other low income countries, particularly in sub-saharan Africa.

Overall, the methods and findings of this study are clear. This study would benefit from some edits to tables so they can be understood without referencing the paper itself and some relatively minor clarifications in-text.

Intro comments

• “since the health of women and children is usually disproportionally affected during pandemics and conflicts” (page 2) I have no doubt this statement is true but was wondering if you could cite evidence for this statement?

• Not mandatory but may be helpful for the reader- “The country has an under-five mortality rate of 56 per 1,000 live births, an infant mortality rate of 42 per 1,000 live births and a neonatal mortality rate of 29 per 1,000 live births [26]. Concerning maternal deaths, the maternal mortality rate is 289 per 100,000 live births [26]” (page 2) do you have any global estimates that you could compare to so the reader doesn’t have to look that up themselves while reading?

• Add when data in table 1 are from to the title of the table (I assume they’re from 2019 since they match numbers from 2019 in introduction section). I suggest to also reference table 1 when speaking about 2019 indicators

• “The COVID-19 pandemic, which was declared a public health emergency in January 2020, reached the Gambia on 17 March 2020” (page 3) I suggest re-wording this to say that COVID-19 was first detected on 17 March 2020 since it may have been circulating in the Gambia before this

• Edits on table 2

1- Suggest re-naming table 2 “COVID-19 Cases in the Gambia Over Time” or something similar so we know cases in your heading are in fact cases of COVID-19

2- Make sure that all headings indicate that these are total #s of cases, deaths, etc.

3- Add a border to separate October to December 2020 and January to March 2021

4- In writing you state “by June 2020, only 48 new cases had been detected” (page 3) but table 2 says there were 45 cases April-June 2020, why are these numbers different?

5- Is this total cases in all of the Gambia or part of the Gambia, and is this all people in the Gambia or just the population of interest. Consider adding a footnote in your table to specify all of this. Is this limited to confirmed cases and is a confirmed case by PCR test only or antigen and PCR here?

• “evidence collected thus far suggests that the uptake of 4 antenatal and immunisation services declined considerably during the first wave of the pandemic” (pages 3-4). When did the first wave end in the Gambia?

Materials and methods comments

•What is the catchment area of these facilities? Are only people from the region of the facility or is it common for people in rural locations and outside of the region to visit one of these hospitals?

• Table 3:

1- Merge the two Kanifing LGA sub-headers so it looks like the Brikama sub-header

2- Center the #s in the respective cells as is done with prior tables.

3- Highlight the second total column. Also suggest to label them differently so they can be easily distinguished to someone reading table without the accompanying methods section.

• “Following this, to ensure the objectivity of the findings, the two researchers worked closely together in reviewing and defining themes” (page 5) so did both reviewers code everything and compare or did each reviewer work on something different and discuss? Please clarify.

Results:

• Results are presented in a cohesive manner- were there any instances where a single health facility or type (e.g. private vs public), or were all of these themes corroborated in all 4 facilities? E.g. was there a lack of PPE and meds in all facililies or just some? Public vs private differences may be noteworthy since public facilities in some low income settings are known to be more under-resourced

Discussion

• From my understanding, another limitation is not knowing how mothers and healthcare workers impressions on accessing MNCH before COVID-19. Some of these issues you discovered may be new, but I am left curious if some existed before the pandemic or were relevant during other outbreaks. I do see you cited Ebola also playing a role in misinformation, but I am interested if Ebola had implications on other factors too… may not be possible to know, so would acknowledge what you don’t know about other outbreaks in limitations.

Conclusions

• “as well as to the need to counter misinformation” (page 13) would also add the need to mitigate fears of the public since this was a big theme of yours.

• Any more specific thoughts on what the facilities and government can do if there’s a future uptick or different pandemic? E.g. maybe recommend things like stockpiling extra PPE, planning for essential med logistics in a pandemic, etc.

6. PLOS authors have the option to publish the peer review history of their article (what does this mean?). If published, this will include your full peer review and any attached files.

Reviewer #1: No

---

## [Author Response · Author response to Decision Letter 0]

28 Mar 2023

Editor Comments:

1. Title: Barriers to accessing mother, new-born and child health services in urban Gambia during COVID-19: An interview-based qualitative study

- Be clear on the objective of the study and be consistent throughout from title till conclusion

MNCH vs antenatal and immunization, these terms are not interchangeable. For one, MNCH includes curative services.

Response: MNCH replaced with antenatal and immunization were applicable

- ‘New-born’ replace with ‘newborn’ throughout

Response: ‘New-born’ replaced with ‘newborn’ throughout the manuscript

- ‘barriers to accessing...’ or ‘factors that influenced utilization of…’

Response: ‘barriers to accessing’ replaced with ‘factors that influenced utilization of’ throughout the manuscript

- “The Gambia” versus “the Gambia” versus “Gambia” – choose one and be consistent

Response: The Gambia used throughout the manuscript 

2. Abstract

- Introduction: Good to avoid terms that are not supported by data “least developed…, least researched…” , low-income country with limited research on ..

Response: “ ….least developed and least researched countries” removed from the abstract

- Results: ‘.. relating to factors that were responsible for the reduction in the uptake of antenatal and immunisation services’ since the study did not measure if there was reduction or not of antenatal and immunization service uptake, a better phrase could be ‘… factors that influenced uptake (use) of antenatal and immunization services...’

Response: ‘.. relating to factors that were responsible for the reduction in the uptake of antenatal and immunisation services’ removed from the abstract.

3. Introduction

4. Methods

Data collection:

- Sample size, why 17 mothers? Did you expect saturation by then? Most were from the 2 hospitals and not so much from the 3rd public hospital and hardly any from the private practice. Was that intended or were you unable to get participants? In either case – this should be reflected in the discussion as limitation, and reasons why.

Response: the initial plan was to recruit as least 24 mothers, 6 from each health facility, but this was not possible due to difficulties in getting participants willing to be interviewed, especially in private health facilities. Most of them did not like the idea of being tape recorded even after telling them it would be totally anonymised. We also decided to include only one private health facility because we realised that the use of MNCH services in private health facilities were hardly disrupted. This limitation has been reflected in the discussion section (page 12, paragraph 5, line 13)

- How did you classify mothers who regularly versus not regularly used services? Were they asked different questions? Were there unique perspective ..? were you trying to see change from their norm in terms of use of service?

Response: mothers who accessed antenatal and immunisation services for at least 4 times during the pandemic were classified as regularly used services, while those who accessed services for less than 4 times were classified as not regularly used services. some of the questions they were asked were similar while some were different. We classified them into these categories so that we can understand the barriers/challenges that those who regularly used services experienced while accessing services and understand the reasons why some mothers did not regularly used services.

- Who conducted the interviews (were the interviewers trained, male/female, health worker/not hw)? Possible social desirability bias given that the interview occurred inside a health facility

Response: All the interviews were conducted by the first author, who was trained on interview techniques by the second author prior to conducting the interviews. This statement has been included in the methods section (page 9, second paragraph). The authors acknowledge that there could be a possible social desirability bias given that the interviews occurred inside a health facility. This statement has been added to the discussion section as a possible limitation (page 12, paragraph 5, line 12).

- How long did the interview take per person?

Response: The interviews lasted between 15-30 minutes. This statement has been included in the methods section (page 5, paragraph 2, line10).

- Were the tool used for interview translated/back translated or? dedicated translator or were the authors/interviewers fluent in all three languages?

Response: The first author is fluent in all the three languages. He conducted all the interviews and translated those conducted in Mandinka and Fula into English. As such, a dedicated translator was not needed. This statement has been included in the methods section (page 5, paragraph 2, line 9).

5. Results

- ‘claimed’ has negative undertone suggest using words like ‘reported’ throughout

Response: ‘claimed’ replaced with ‘reported’ throughout the manuscript

- ‘Fear of being quarantined’ is this similar to being asked to stay at home and to social distance. Good to differentiate and clearly state.

Response: they are different. At the start of the pandemic, when someone tests positive, they would be taken to a government facility, usually a hotel, where they will stay at least for 14 days before they will be allowed to go home. This was later changed to allow those who test positive to self-isolate at home for 14 days. Changes made to the manuscript (page 7, paragraph 2, line 5)

- Given your small sample size per health facility, it is possible that the respondents might be identified by health facility name. We often advise to not include name of hospital or say hospital 1 or 2 to avoid possible identification of interviewee and breach of confidentiality.

Response: Hospital names removed from all quotes, only numbers used.

- Some of the interviewees were describing what they perceive was going on in their community – good to say that in your discussion section some of your findings were also the interviewees perspective of what was going on in their community and not a personal experience.

o “There were several people in my community who stopped taking their children for immunisation during the pandemic..”.

Response: This statement has been included in the discussion section (page 12, paragraph 5, line 15)

- Reference “WhatsApp”

Response: “WhatsApp” removed from the manuscript

- “misleading information” or “rumors” – trying to understanding if the respondents defined the information as “misleading’ or “rumor” ? as some of the health facilities were closed as described under paragraph Scaling down of MNCH services.

Response: both terms were used by participants to define the information. 

6. Discussion

- Hard to follow flow of this section. Suggest restructuring: main/key finding, interpretation of your findings/reflection of findings as it relates to other African and non-African studies, implications of the findings, strength/limitations of study, recommendation (what could be done differently in the future based on your findings).

Response: Discussion section restructured based on the above suggestions.

- First sentence

o Given the small sample and lack of generalizability in Gambia suggest editing to say gives insight in Urban areas of Gambia.

Response: Wording of the first sentence changed to “urban areas of The Gambia”

- Second sentence

o ‘Factors responsible for reduction’… suggest rewording… the study does not measure reduction but rather factors that influenced decision to use or not use services.

Response: ‘Factors responsible for reduction’ replaced with ‘factors that influenced the use’

- Line 8

o “physical barriers” .. suggest to say barriers

Response: “physical barriers’ replaced with “barriers”

- ‘small” country as compared to ?

Response: “small” removed from the manuscript

- Limitations: I would also think these were limitations

o The small sample size , unclear if saturation was researched

o The sampling method, selection bias as the pool of patients was drawn from hospital and does not include those who did not seek care at all. The authors acknowledge that there could be a possible social desirability bias given that the interviews occurred inside a health facility.

o Urban centric as sample was from urban areas.

o Hence, findings not generalizable to Gambia but provides insight for the country.

Response: these points have been included in the limitations (page 12, paragraph 5, line 10 – 19).

7. Figure 1. Missing title of Figure. Not sure it adds value. Is it trying to capture framework or key findings and needs language edits “mistreatment by ..”

Response: Figure 1. removed from the manuscript.

Reviewer Comments to the Author:

1. Intro comments

• “since the health of women and children is usually disproportionally affected during pandemics and conflicts” (page 2) I have no doubt this statement is true but was wondering if you could cite evidence for this statement?

Response: We thank the above reviewer for highlighting this lack of clarity. We have decided to remove the above statement from the manuscript.

• Not mandatory but may be helpful for the reader- “The country has an under-five mortality rate of 56 per 1,000 live births, an infant mortality rate of 42 per 1,000 live births and a neonatal mortality rate of 29 per 1,000 live births [26]. Concerning maternal deaths, the maternal mortality rate is 289 per 100,000 live births [26]” (page 2) do you have any global estimates that you could compare to so the reader doesn’t have to look that up themselves while reading?

Response: Thank you for this suggestion. Here are some global estimates: global maternal mortality rate: 223 per 100,000 live births, under-5 mortality rate: 38 per 1,000 live births, infant mortality rate: 29 per 1,000 live births, neonatal mortality rate of 18 per 1,000 live births.

• Add when data in table 1 are from to the title of the table (I assume they’re from 2019 since they match numbers from 2019 in introduction section). I suggest to also reference table 1 when speaking about 2019 indicators

Response: Thank you for this additional reference. 2019 added to title of table 1 and table 1 referenced when speaking about 2019 indicators (page 2, paragraph 4 and 5).

• “The COVID-19 pandemic, which was declared a public health emergency in January 2020, reached the Gambia on 17 March 2020” (page 3) I suggest re-wording this to say that COVID-19 was first detected on 17 March 2020 since it may have been circulating in the Gambia before this

Response: Thank you for this suggestion. The wording of the above sentence changed to “The first case of COVID-19 was detected in The Gambia on 17 March 2020”.

• Edits on table 2

1- Suggest re-naming table 2 “COVID-19 Cases in the Gambia Over Time” or something similar so we know cases in your heading are in fact cases of COVID-19

Response: Thank you for this suggestion. Title of table 2 renamed as ““COVID-19 Cases in the Gambia Over Time” (page 3)

2- Make sure that all headings indicate that these are total #s of cases, deaths, etc.

Response: Thank you for this suggestion. Changes made as suggested.

3- Add a border to separate October to December 2020 and January to March 2021.

Response: We are a bit unsure about this recommendation, as you will notice there is already a border in the table. Please advise on further action.

4- In writing you state “by June 2020, only 48 new cases had been detected” (page 3) but table 2 says there were 45 cases April-June 2020, why are these numbers different?

Response: Thank you for raising this. 45 news cases between April and June + 3 new cases in March excluding the first case = 48 new cases by end of June excluding the first case. Wording has been changed to “45 new cases by the end of June” (page 3, paragraph 1, line 4).

5- Is this total cases in all of the Gambia or part of the Gambia, and is this all people in the Gambia or just the population of interest. Consider adding a footnote in your table to specify all of this. Is this limited to confirmed cases and is a confirmed case by PCR test only or antigen and PCR here?

Response: Thank you for highlighting this lack of clarity. It is the total cases in all of The Gambia. It is also all the people in The Gambia not just the population of interest. The data is limited to confirmed cases only. I am not sure how cases were confirmed but all the data used here are from the Ministry of Health reports. This clarification has now been added in Table 2 and in page 3, paragraph 1).

• “evidence collected thus far suggests that the uptake of 4 antenatal and immunisation services declined considerably during the first wave of the pandemic” (pages 3-4). When did the first wave end in the Gambia?

Response: Thank you for this question. The first wave ended in September 2020. This clarification has now been added in page 4, paragraph 1, line 4).

2. Materials and methods comments

•What is the catchment area of these facilities? Are only people from the region of the facility or is it common for people in rural locations and outside of the region to visit one of these hospitals?

Response: Thank you for these questions. The health facilities are from the two most densely populated local government areas in the county, representing about 57% of the country’s population. With regards to antenatal and immunization services, only patients living in these areas access such services. This clarification has now been added in page 4, paragraph 4, line 2.

• Table 3:

1- Merge the two Kanifing LGA sub-headers so it looks like the Brikama sub-header

Response: Thank you for this suggestion. Merged the two Kanifing LGA sub-headers 

2- Center the #s in the respective cells as is done with prior tables.

Response: Thank you for this suggestion. Centred the numbers in the respective cells

3- Highlight the second total column. Also suggest to label them differently so they can be easily distinguished to someone reading table without the accompanying methods section.

Response: Thank you for this suggestion. Changes made as suggested.

• “Following this, to ensure the objectivity of the findings, the two researchers worked closely together in reviewing and defining themes” (page 5) so did both reviewers code everything and compare or did each reviewer work on something different and discuss? Please clarify.

Response: Thank you for highlighting this lack of clarity. Both researchers coded everything and compare. This statement has been included in the data analysis section ( page 6, paragraph 1, line 9)

3. Results: 

• Results are presented in a cohesive manner- were there any instances where a single health facility or type (e.g. private vs public), or were all of these themes corroborated in all 4 facilities? E.g. was there a lack of PPE and meds in all facililies or just some? Public vs private differences may be noteworthy since public facilities in some low income settings are known to be more under-resourced

Response: Thank you for raising this. For some of the themes, they were corroborated in all 4 facilities, but that is not the case for every theme. When doing the analysis, we tried to find the most common themes among all the facilities, but that was not always possible since most of our participants, particularly the mothers, were drawn from only two health facilities. Public vs private differences were difficult to determine since we had only 3 participants from the private sector, but still, we noticed that the disruption on MNCH services was more common in the public sector compared to the private sector.

4. Discussion

• From my understanding, another limitation is not knowing how mothers and healthcare workers impressions on accessing MNCH before COVID-19. Some of these issues you discovered may be new, but I am left curious if some existed before the pandemic or were relevant during other outbreaks. I do see you cited Ebola also playing a role in misinformation, but I am interested if Ebola had implications on other factors too… may not be possible to know, so would acknowledge what you don’t know about other outbreaks in limitations.

Response: Thank you for this suggestion. We understand that some of the factors highlighted by study participants existed even before the pandemic, but the participants highlighted that some of these factors got worse during the pandemic, such as lack of medications. With regards to the implication of Ebola, we don’t really now, so we have now added this as a limitation (page 12, paragraph 5, line 3)

5. Conclusions

• “as well as to the need to counter misinformation” (page 13) would also add the need to mitigate fears of the public since this was a big theme of yours.

Response: Thank you for this suggestion. “the need to mitigate fears of the public” added to page 13, paragraph 3, line 4.

• Any more specific thoughts on what the facilities and government can do if there’s a future uptick or different pandemic? E.g. maybe recommend things like stockpiling extra PPE, planning for essential med logistics in a pandemic, etc.

Response: Thank you for this suggestion. More recommendations added based on the above suggestions (page 13, paragraph 3, line 4)

---

## [Decision Letter · Decision Letter 1]

26 Apr 2023

PONE-D-22-27435R1

Factors that influenced utilization of antenatal and immunization services in two local government areas in The Gambia during COVID-19: An interview-based qualitative study.

PLOS ONE

Dear Dr. Bah,

Thank you for submitting your manuscript to PLOS ONE. After careful consideration, we feel that it has merit but does not fully meet PLOS ONE’s publication criteria as it currently stands. Therefore, we invite you to submit a revised version of the manuscript that addresses the points raised during the review process.

We have added few comments to improve the manuscript and meet PLOS ONE’s publication criteria.

We look forward to receiving your revised manuscript.

Kind regards,

Ribka Amsalu

Academic Editor

PLOS ONE

Journal Requirements:

Additional Editor Comments:

There are minor grammatical errors and few more comments/suggestions.

1. Introduction section

Keep the mortality information as it is relevant for readers and makes a point that in countries that have high infant, child and maternal mortality the indirect impact of the pandemic can be severe. Minor edit - it is Maternal mortality ratio (not rate). 

2. Discussion section

"Temesgan and colleagues [14] found that mothers who were not required to obtain permission from their partners to access antenatal and immunization services during the pandemic had greater odds of accessing antenatal and immunization services than those who were required to seek permission" While an interesting finding I was not clear how the findings relate to the results of your study. 

Reviewers' comments:

Reviewer's Responses to Questions

**Comments to the Author**

1. If the authors have adequately addressed your comments raised in a previous round of review and you feel that this manuscript is now acceptable for publication, you may indicate that here to bypass the “Comments to the Author” section, enter your conflict of interest statement in the “Confidential to Editor” section, and submit your "Accept" recommendation.

Reviewer #1: All comments have been addressed

2. Is the manuscript technically sound, and do the data support the conclusions?

Reviewer #1: Yes

3. Has the statistical analysis been performed appropriately and rigorously? 

Reviewer #1: N/A

4. Have the authors made all data underlying the findings in their manuscript fully available?

Reviewer #1: Yes

5. Is the manuscript presented in an intelligible fashion and written in standard English?

Reviewer #1: Yes

6. Review Comments to the Author

Reviewer #1: Topline comment:

• Check throughout paper that you are using plural verbs when referring to data, since you have more than one piece of data (i.e. use “data were” rather than “data was”)

Abstract

“we identified five main themes, ranging from individual factors to interpersonal, community, institutional and policy factors”

• Seems you identified more than five themes since each of these areas has more than one theme, suggest to instead say “we identified themes at five different levels: individual, interpersonal, etc” or something along this line for clarity.

Intro

“With the recent COVID-19 pandemic, the achievements made globally in improving mother”

• This statement implies that the pandemic is over, but there are still cases occurring globally,” consider rewording to “ongoing COVID-19 pandemic” or “COVID pandemic declared as a public health emergency on x date” or something similar.

“During the early phase of the pandemic there was a concern about the potential”

• very minor but would add comma after the word pandemic.

“The demands-side factors include movement restrictions, economic hardship”

• Minor but take out the s in demands here to keep language consistent

“However, despite these achievements in increased MNCH uptake, when compared to international rates, the maternal, newborn and child mortality rates in The Gambia remain very high”

• Suggest to remove the words “when compared to international rates” since you didn’t provide international rates in relation to MNCH uptake, this will leave the reader wanting international rates to compare to.

• You talk about MNCH and how it has improved over time, but then for mortality rates, you don’t mention change over time… is this because mortality hasn’t improved over time? Instead of focusing on mortality rates being high and not saying what they are high in comparison to (presumably it’s global rates but you don’t explicitly say this or what those global rates are), would focus on this rate not improving if this is true. I also recommend to consider why you even included mortality information in this paper if the focus is antenatal and immunization services and threats to improvement of this. Suggest to add a sentence to drive home why it matters in the context of this paper or delete this paragraph.

“On 27 March, the country declared a state of emergency, which included the closing of schools, non-essential shops, and places of worship, the prohibition of social gatherings of more than 10 people, and the limiting of the number of passengers on public transport”

• Did the state of emergency mandate these things? Suggest to change verb if this is what is meant.

“Between April and July 2020, the government introduced contact tracing and quarantine measures for suspected and confirmed cases, who were obliged to remain in hotels for 14 days.”

• So quarantine meant they were obliged to stay in a hotel? If this is what you meant, suggest to reword to “and quarantine/isolation measures for suspected and confirmed cases which obliged cases to remain in hotels for 14 days” And what about contacts… did they need to quarantine in a hotel?

• Also, isolation applies to cases to prevent spread of illness and quarantine applies to contacts who are not yet ill, so I suggest to check that you use these terms correctly throughout discussion on quarantine/isolation.

“To ensure confidentiality and anonymity, the interview data was anonymised.”

• The use of anonymous twice is redundant, maybe say data were de-identified before analysis. Also make sure you refer to data as plural rather than singular here and throughout the paper

“They were reported to have the highest prevalence of COVID-19 cases in the country [43].”

• Just looked at this citation and didn’t see anything about prevalence of COVID-19, seems like you may have cited the wrong source.

My prior comments on methods, results, discussion, conclusion have been resolved and I have no further comments at this time

7. PLOS authors have the option to publish the peer review history of their article (what does this mean?). If published, this will include your full peer review and any attached files.

Reviewer #1: No

---

## [Author Response · Author response to Decision Letter 1]

29 May 2023

Dear Editors,

Editor Comments:

1. 1. Introduction section

Keep the mortality information as it is relevant for readers and makes a point that in countries that have high infant, child and maternal mortality the indirect impact of the pandemic can be severe. Minor edit - it is Maternal mortality ratio (not rate). 

Response: We have kept the mortality information as suggested and have changed Maternal mortality rate to maternal mortality ratio.

2. Discussion section

"Temesgan and colleagues [14] found that mothers who were not required to obtain permission from their partners to access antenatal and immunization services during the pandemic had greater odds of accessing antenatal and immunization services than those who were required to seek permission" While an interesting finding I was not clear how the findings relate to the results of your study.

Response: the above citation has been removed from the transcript and replaced with: “Hailemariam and colleagues [18] found that mothers were unwilling to seek antenatal services due to fear of mistreatment by health workers and perceived poor quality of care arising from lack of motivation of health workers to provide antenatal services”.

Reviewer Comments to the Author:

1. Check throughout paper that you are using plural verbs when referring to data, since you have more than one piece of data (i.e. use “data were” rather than “data was”)

Response: Thank you for this suggestion. “Data was” replaced with “data were” throughout the article.

Abstract

2. “we identified five main themes, ranging from individual factors to interpersonal, community, institutional and policy factors”

• Seems you identified more than five themes since each of these areas has more than one theme, suggest to instead say “we identified themes at five different levels: individual, interpersonal, etc” or something along this line for clarity.

Response: Thank you for highlighting this lack of clarity. Changes made as suggested (page 1, paragraph 3, line 1)

Intro

3. “With the recent COVID-19 pandemic, the achievements made globally in improving mother”

• This statement implies that the pandemic is over, but there are still cases occurring globally,” consider rewording to “ongoing COVID-19 pandemic” or “COVID pandemic declared as a public health emergency on x date” or something similar.

Response: Thank you for this suggestion. “With the recent COVID-19 pandemic” replaced with “ongoing COVID-19 pandemic” (page 2, paragraph 1, line 1)

4. “During the early phase of the pandemic there was a concern about the potential”

• very minor but would add comma after the word pandemic.

Response: Thank you for this suggestion. Comma added as suggested (page 2, paragraph 1, line 3).

5. “The demands-side factors include movement restrictions, economic hardship”

• Minor but take out the s in demands here to keep language consistent.

Response: Thank you for this suggestion. The s in demands removed as suggested (page 2, paragraph 3, line 4)

6. “However, despite these achievements in increased MNCH uptake, when compared to international rates, the maternal, newborn and child mortality rates in The Gambia remain very high”

• Suggest to remove the words “when compared to international rates” since you didn’t provide international rates in relation to MNCH uptake, this will leave the reader wanting international rates to compare to.

Response: Thank you for this suggestion. “When compared to international rates” removed as suggested.

7. You talk about MNCH and how it has improved over time, but then for mortality rates, you don’t mention change over time… is this because mortality hasn’t improved over time? Instead of focusing on mortality rates being high and not saying what they are high in comparison to (presumably it’s global rates but you don’t explicitly say this or what those global rates are), would focus on this rate not improving if this is true. I also recommend to consider why you even included mortality information in this paper if the focus is antenatal and immunization services. and threats to improvement of this. Suggest to add a sentence to drive home why it matters in the context of this paper or delete this paragraph.

Response: We appreciate your suggestion, but we have decided to keep the mortality information as the editor suggested.

8. “On 27 March, the country declared a state of emergency, which included the closing of schools, non-essential shops, and places of worship, the prohibition of social gatherings of more than 10 people, and the limiting of the number of passengers on public transport”

• Did the state of emergency mandate these things? Suggest to change verb if this is what is meant.

Response: Thank you for this suggestion. Verb changed to mandated (page 3, paragraph 2).

9. “Between April and July 2020, the government introduced contact tracing and quarantine measures for suspected and confirmed cases, who were obliged to remain in hotels for 14 days.”

• So quarantine meant they were obliged to stay in a hotel? If this is what you meant, suggest to reword to “and quarantine/isolation measures for suspected and confirmed cases which obliged cases to remain in hotels for 14 days” And what about contacts… did they need to quarantine in a hotel?

Response: Thank you for asking for clarification. Contacts were initially required to quarantine for 14 days, but later, quarantine in hotels was replaced with self-isolation at home for both suspected and confirmed cases.

10. Also, isolation applies to cases to prevent spread of illness and quarantine applies to contacts who are not yet ill, so I suggest to check that you use these terms correctly throughout discussion on quarantine/isolation.

Response: Thank you for this suggestion. We have made the following changes: “Between April and July 2020, the government introduced contact tracing and quarantine measures which obliged suspected and confirmed cases to remain in hotels for 14 days. As the pandemic progressed, hotel quarantine was replaced with self-isolation at home for both suspected and confirmed cases for a period of 10” (page 3, paragraph 2, line 7-9). With regard to your question about quarantine of contacts, 

11. “To ensure confidentiality and anonymity, the interview data was anonymised.”

• The use of anonymous twice is redundant, maybe say data were de-identified before analysis. Also make sure you refer to data as plural rather than singular here and throughout the paper.

Response: Thank you for this suggestion. “The interview data was anonymised” replaced with “data were de-identified before analysis” (page 4, paragraph 3)

12. “They were reported to have the highest prevalence of COVID-19 cases in the country [43].”

• Just looked at this citation and didn’t see anything about prevalence of COVID-19, seems like you may have cited the wrong source.

Response: Thank you for bringing this to our attention. Our sincere apologies, the source reported the incidence not the prevalence in the second paragraph of its methods section. We have made the following change: “They were reported to have the highest incidence of confirmed COVID-19 cases in the country”.

---

## [Editor Report · Decision Letter 2]

8 Jun 2023

Factors that influenced utilization of antenatal and immunization services in two local government areas in The Gambia during COVID-19: An interview-based qualitative study.

PONE-D-22-27435R2

Dear Dr. Abdourahman Bah

We’re pleased to inform you that your manuscript has been judged scientifically suitable for publication and will be formally accepted for publication once it meets all outstanding technical requirements.

Kind regards,

Dr. Ribka Amsalu

Academic Editor

PLOS ONE

---

## [Editor Report · Acceptance letter]

15 Jun 2023

PONE-D-22-27435R2 

Factors that influenced utilization of antenatal and immunization services in two local government areas in The Gambia during COVID-19: An interview-based qualitative study. 

Dear Dr. Bah:

I'm pleased to inform you that your manuscript has been deemed suitable for publication in PLOS ONE. Congratulations! Your manuscript is now with our production department. 

Kind regards, 

on behalf of

Dr. Ribka Amsalu 

Academic Editor

PLOS ONE